# Machine learning prediction of non-attendance to postpartum glucose screening and subsequent risk of type 2 diabetes following gestational diabetes

**Nishanthi Periyathambi**[1,2]☯, **Durga Parkhi**[1]☯, **Yonas Ghebremichael-Weldeselassie**[1,3‡], **Vinod Patel**[2‡], **Nithya Sukumar**[1,2‡], **Rahul Siddharthan**[4,5‡], **Leelavati Narlikar**[6¤‡], **Ponnusamy Saravanan**[1,2]*

**1** Division of Populations, Evidence, and Technologies of Health Sciences, Warwick Medical School, University of Warwick, Coventry, United Kingdom, **2** Department of Diabetes, Endocrinology, and Metabolism, George Eliot Hospital, Nuneaton, United Kingdom, **3** School of Mathematics and Statistics, The Open University, Milton Keynes, United Kingdom, **4** Department of Computational Biology, The Institute of Mathematical Sciences, Chennai, India, **5** Homi Bhabha National Institute, Mumbai, India, **6** Department of Chemical Engineering, CSIR-National Chemical Laboratory, Pune, India

☯ These authors contributed equally to this work.
¤ Current address: Department of Data Science, Indian Institute of Science Education and Research, Pune, India
‡ YGW, VP, NS, RS and LN also contributed equally to this work.
* P.Saravanan@warwick.ac.uk

## Abstract

### Objective

The aim of the present study was to identify the factors associated with non-attendance of immediate postpartum glucose test using a machine learning algorithm following gestational diabetes mellitus (GDM) pregnancy.

### Method

A retrospective cohort study of all GDM women (n = 607) for postpartum glucose test due between January 2016 and December 2019 at the George Eliot Hospital NHS Trust, UK.

### Results

Sixty-five percent of women attended postpartum glucose test. Type 2 diabetes was diagnosed in 2.8% and 21.6% had persistent dysglycaemia at 6–13 weeks post-delivery. Those who did not attend postpartum glucose test seem to be younger, multiparous, obese, and continued to smoke during pregnancy. They also had higher fasting glucose at antenatal oral glucose tolerance test. Our machine learning algorithm predicted postpartum glucose non-attendance with an area under the receiver operating characteristic curve of 0.72. The model could achieve a sensitivity of 70% with 66% specificity at a risk score threshold of 0.46. A total of 233 (38.4%) women attended subsequent glucose test at least once within the first two years of delivery and 24% had dysglycaemia. Compared to women who

**Data Availability Statement:** All relevant data are within the paper and its Supporting Information files.

**Funding:** PS and YW are part funded by Medical Research Council, UK Grant number: MR/R020981/1 Funder: MRC- UK URL: https://mrc.ukri.org/funding/. NP is funded by Chancellor's international Scholarship for doctoral research. Funder: University of Warwick URL: https://warwick.ac.uk/services/dc/schols_fund/scholarships_and_funding/chancellors_int/. RS is supported by his institution, funded by the Department of Atomic Energy, Government of India. We declare that the funders had no role in study design, data collection and analysis, decision to publish, or preparation of the manuscript and, there was no additional external funding received for this study.

**Competing interests:** The authors have declared that no competing interests exist.

attended postpartum glucose test, those who did not attend had higher conversion rate to type 2 diabetes (2.5% vs 11.4%; p = 0.005).

## Conclusion

Postpartum screening following GDM is still poor. Women who did not attend postpartum screening appear to have higher metabolic risk and higher conversion to type 2 diabetes by two years post-delivery. Machine learning model can predict women who are unlikely to attend postpartum glucose test using simple antenatal factors. Enhanced, personalised education of these women may improve postpartum glucose screening.

## Introduction

Gestational diabetes mellitus (GDM) is associated with adverse maternal and offspring outcomes both in the short and long-term. Incidence of type 2 diabetes (T2D) in women with history of GDM using real-world data can be 20-fold higher [1]. The conversion to T2D following the pregnancy seems to happen early in the postpartum period with highest risk within 3–6 years of index pregnancy; by 10–14 years, 50% of women have dysglycaemia [2, 3]. GDM is also associated with at least two-fold greater risk for developing hypertension and cardiovascular disease (CVD) [1, 4]. This highlights the importance of early identification of women at risk to implement any preventive strategies [5, 6]. Most international guidelines recommend 75-g oral glucose tolerance test (OGTT) or HbA1c for all GDM women in the postpartum period, and then at least once every 1–3 years [7, 8].

In 2015, the National Institute of Health and Care Excellence (NICE) changed the screening guidelines from every 3 years to every year and recommended a fasting plasma glucose (FPG) test at 6 to 13 weeks after delivery or HbA1c at 13 weeks onwards instead of OGTT [9]. Despite these recommendations and evidence of diabetes risk, postpartum testing for dysglycaemia (by any form of glucose testing; OGTT, FPG or HbA1c) is low and reaches only about 30% by year three following a GDM pregnancy [1, 10]. While some centres were able to improve their uptake to 70% by dedicated coordinators, this has not been replicated by others and uptake remained poor [11, 12]. System and patient barriers including the lack of awareness ('it does not affect me'), discomfort and duration of the test (especially for OGTT), other socio- economic factors and poor transition of information from secondary to primary care are primary reasons for poor uptake of postpartum screening [13, 14].

We had earlier shown that women who did not attend postpartum glucose testing (ppGT) in three centres from our region had higher metabolic risk factors [10]. Subsequently, in one of the centres, we introduced a dedicated advanced nurse practitioner to encourage women to attend the postpartum testing and send personalized letters to women and their GPs for annual screening. However, there has been limited data on an individualised approach for targeting women who are unlikely to attend the ppGT. The primary aim of this study was to identify patient characteristics of women who did not attend the immediate ppGT in a real-world setting and to assess their subsequent T2D risk within 24-months post-delivery. Further, we built a predictive model to identify women who are less likely to attend ppGT using machine learning.

## Methods

### Study population

Detailed demographic, clinical and anthropometric data for all pregnant women who had GDM and ppGT due between January 2016 and December 2019 ($n$ = 607) in a district general hospital (George Eliot Hospital NHS Trust, Nuneaton, UK) were collected. No personal information of the subjects was obtained. This study was conducted as a service evaluation audit and ethical approval was not necessary. The audit was approved by the GEH Diabetes and Audit department. A selective screening was done based on NICE 2015 criteria, using 75g OGTT between 24 and 28 weeks of gestation. GDM diagnosis was made if FPG ≥5.6 mmol/L or 2- hour plasma glucose ≥7.8 mmol/L. The risk assessment for screening includes women with higher pre-pregnancy BMI (≥30 kg/m$^2$), family history of diabetes, previous GDM, previous macrosomic baby (birth weight ≥ 4500g) and from ethnic minority groups [9]. At the time of discharge, all GDM women were instructed to schedule a ppGT at 6 to 13 weeks and received a phone reminder in the previous week of appointment by an advanced nurse practitioner.

In addition to the detailed antenatal data, data at birth including initiation of breastfeeding and birth centile of the infant, assessed by ethnicity specific UK reference chart were collected [15]. Women received a personalized letter to attend the ppGT (OGTT for those who had normal FPG and abnormal 2-hr glucose during antenatal OGTT or HbA1c for other abnormalities) (S1 Appendix). If they had difficulty in attending OGTT, they were invited again for postpartum HbA1c test. Following the results, GDM women and their General Practitioner (GP) received a second letter indicating their results and highlighting their future risk of T2D and CVD and advising them to visit their GP for annual HbA1c tests. All the follow-up glucose/HbA1c data (up to 24 months from the date of delivery) were extracted from the hospital electronic database. During follow-up, dysglycaemia (diabetes and prediabetes) was identified using American Diabetes Association (ADA) criteria [diabetes: HbA1c ≥48 mmol/mol (≥ 6.5%) or FPG ≥7.0mmol/L and 2-hr ≥11.1mmol/L post 75g OGTT; prediabetes: HbA1c ≥39 and <48 mmol/mol (≥5.7 and 6.4%) or FPG ≥5.6 and ≤6.9mmol/L and 2-hr ≥7.8 to ≤11.0 mmol/L post 75g OGTT] [16].

### Statistical analysis

Statistical analysis was performed using IBM SPSS Statistics for Windows, version 27 (IBM Corp., Armonk, NY, USA). Baseline characteristics were expressed in percentages for categorical variables and mean ± standard deviation (SD) for continuous variables. Univariate and multivariate hazard ratios (HR) were estimated using Cox proportional hazard model in a subgroup of women who were followed for two years from the date of delivery. All the potential predictors of dysglycaemia during the two years follow-up period were adjusted in the final model (S1 Table).

### Machine learning analysis

Machine Learning (ML) analysis was performed in Python version 3.7 (www.python.org). Lasso regularization was used for feature selection. Nested standard 5-fold cross validation (CV1) was used for model evaluation [17]. An internal stratified 10-fold cross validation (CV2) was performed on each of the five training folds of CV1 for optimizing the shrinkage parameter in lasso (S1 Fig). Missing values were imputed using Multivariate Imputation by Chained Equations (MICE) technique, using other non-missing covariates, separately for the training and testing folds of CV1 to avoid leakage of information from the testing data into the

training data. The training folds in CV1 were resampled using adaptive synthetic resampling technique to ensure equal representation of both the binary classes. The resampled training data was normalized.

Logistic regression model was fitted on the training folds in CV1 using the selected features from lasso. The model predictions on each of the test folds in 5 iterations of CV1 were aggregated and the Receiver Operating Characteristic (ROC) curve was plotted for this aggregated set. The area under the ROC curve (AUROC) was used to assess the performance of the method. The concept diagram of this method is illustrated in S2 Fig. After getting assurance of acceptable performance of this method, logistic regression was applied stepwise on the full data to obtain the final model for deployment. Stepwise details of our proposed method are given below:

Step 1: Lasso hyperparameter optimization–Lasso regularization embeds feature selection in the form of an L1 penalty on the magnitude of the feature coefficients, causing irrelevant feature coefficients to shrink to zero. The shrinkage hyperparameter, i.e., the magnitude of the penalty, is optimized using a stratified 10-fold cross validation (CV2) on the training folds of CV1 in each iteration $i$. This process gives us an optimal hyperparameter value $C_i$ for each fold $i$.

Step 2: Feature selection- All baseline features (except history of alcohol in current pregnancy and previous pregnancy GDM history) were considered for feature selection. Converting the categorical features to binary using one-hot-encoding, there were 27 features in total. Lasso with the tuned hyperparameter $C_i$ from step 1 was used to select the best feature set $f_i$ from the training data in iteration $i$.

Step 3: Model training- Logistic regression model $m_i$ was learned from the features $f_i$, selected in step 2, for each fold $i$ of CV1.

Step 4: Model evaluation- The logistic regression model $m_i$ in step 3 was used to make predictions on the held-out test data of the corresponding fold $i$ of CV1. The predictions on each of the 5 test folds of CV1 were aggregated to plot the ROC and calculate the AUROC.

Finally, Steps 1 to 3 were applied on the full data to obtain the final model for practical use. That is, lasso regularization hyperparameter was optimized using stratified 10-fold cross validation on the full data, features were selected from the full data using lasso with optimized $C$, and logistic regression model was learned on the selected features of the full data.

Analysis of Specificity, Positive Predictive Value (PPV), Negative Predictive Value (NPV), F1-score, and Accuracy was performed for five predetermined values of Sensitivity (60%, 70%, 75%, 80%, and 90%) for the optimal selected model. Using the coefficients from the final fitted logistic regression model on the full data, a composite risk scoring system was developed using the best selected antenatal predictors to predict the probability of GDM women to miss ppGT. Composite risk score was calculated from the equation, $1/[1+\exp(-b)]$, where $b = b_0 + b_1 x_1 + b_2 x_2 + \ldots + b_n x_n$ where $b_0$ is the intercept and $b_n$ coefficient of $n^{\text{th}}$ predictor ($x_n$), respectively. A Decision Curve Analysis (DCA) was used to evaluate and compare the performance of our model in comparison to 'target all' and 'target none' approaches [18]. Finally, the correctly identified non-attenders (sensitivity) vs follow-ups avoided (the true negatives + false negatives, obtained from the optimal selected model) were used to calculate the number of women requiring enhanced care, to maximize the postpartum follow-up care.

## Results

In total, 607 pregnant women were diagnosed with GDM. Diagnosis of GDM was made at mean gestational week of 27.9±4.4 weeks. 7.4% ($n$ = 45) of women were diagnosed by the FPG

alone, 58.3% (*n* = 354) by 2-hrs glucose alone and 12.4% (*n* = 75) had abnormal values for both. Further, 21.9% (*n* = 133) GDM diagnosed women had missing data on antenatal OGTT which were imputed using MICE technique. The prevalence of large for gestational age (LGA) was 13.8% and small for gestational age (SGA) was 14.7%.

Overall, 64.9% (*n* = 394) GDM women attended the ppGT, including postnatal OGTT (*n* = 340) and HbA1c (*n* = 54). Subsequently, 233 women had follow-up glucose testing at least once within 24-months of index delivery. At the immediate ppGT, 21.6% had dysglycaemia/impaired glucose regulation (abnormal impaired fasting glucose (IFG), impaired glucose tolerance (IGT) or HbA1c ≥39 and <48 mmol/mol) and 2.8% had been diagnosed with T2D (Fig 1). Comparison of the anthropometric, clinical, and demographic characteristics of women who did and did not attend the ppGT is shown in Table 1.

Women who did not attend ppGT were younger, unmarried, multiparous, had higher BMI and continued to smoke during pregnancy at the antenatal booking visit compared to those who attended ppGT (Table 1). The factors independently associated with non-attendance of ppGT are shown in Table 2. The factors associated with non-attendance of ppGT, selected in each of the 5 iterations of cross validation are shown in S3 Fig.

The AUROC computed from aggregating the test predictions from 5 test folds is 0.72 (Fig 2). The optimal threshold was determined as 0.46 (sensitivity of 0.70, specificity of 0.66 and, maximal F1 score). Forty six out of 100 women were above this threshold of 0.46 and focusing on these women could improve the ppGT testing (S4 Fig). Table 3 shows the sensitivity, specificity, PPV, NPV, F1 score and accuracy at other probability thresholds. The F1 graph is shown in S5 Fig.

In the decision curve analysis, by comparing the 'target all' and 'target none' approaches, ML algorithm obtained higher standardized net benefit as compared to the follow-up of all

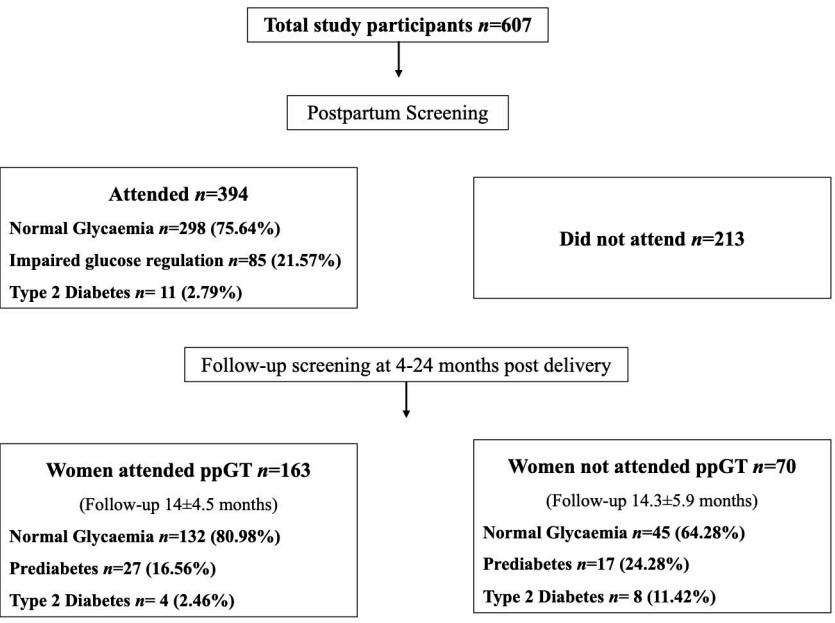

**Fig 1. Consort diagram of immediate postpartum glucose testing attendance and follow-up for 24-months post-delivery.** The flow chart displayed the proportion of GDM women attended vs not attended ppGT. The diagnosis of dysglycaemia/impaired glucose regulation and Type 2 diabetes was performed as, Normal glycaemia: FPG <5.6 and 2-hr glucose <7.8 at postpartum OGTT or HbA1c <39 mmol/mol (<5.7%); Impaired glucose regulation: IFG, IGT and/or Prediabetes [Impaired Fasting glucose: FPG ≥5.6mmol/L; Impaired Glucose tolerance: 2-hr glucose ≥7.8 mmol/L; Prediabetes: HbA1c ≥39 to <48 mmol/mol (≥5.7 and <6.4%)]; Type 2 diabetes: FPG ≥7.0mmol/L and/or 2-hr ≥11.1mmol/L post 75g OGTT or HbA1c ≥48mmol/mol (≥6.5%).

**Table 1. Antenatal, delivery and postnatal characteristics of GDM women with postpartum glucose screening attendance.**

| Characteristics | All women $n$ = 607 | Attended $n$ = 394 | Did not attend $n$ = 213 | p-value |
|---|---|---|---|---|
| **At booking** | | | | |
| Age** | 31.59±5.76 | 32.21±5.40 | 30.45±6.22 | 0.001 |
| Parity*** | 1.98±1.17 | 1.78±0.98 | 2.35±1.38 | <0.001 |
| Multiparous (≥2)** | 339/604 (56.1%) | 200/392 (51%) | 139/212 (65.6%) | 0.001 |
| Weight (Kg)** | 81.86±20.51 | 79.78±19.79 | 85.58±21.27 | 0.001 |
| Height (m) | 1.64±0.07 | 1.64±0.07 | 1.64±0.08 | 0.613 |
| BMI (Kg/m$^2$)** | 30.49±7.14 | 29.76±6.81 | 31.79±7.53 | 0.001 |
| • < 18.5 | 10/557 (1.8%) | 6/357 (1.7%) | 4/200 (2.0%) | 0.004 |
| • 18.5 to 24.9 | 125/557 (22.4%) | 92/357 (25.8%) | 33/200 (16.5%) | |
| • 25 to 29.9 | 141/557 (25.3%) | 99/357 (27.7%) | 42/200 (21.0%) | |
| • ≥30 | 281/557 (50.4%) | 160/357 (44.8%) | 121/200 (60.5%) | |
| **Blood Pressure (BP; mmHg)** | | | | |
| • Systolic BP | 115.8±13.23 | 115.71±13.62 | 115.97±12.54 | 0.827 |
| • Diastolic BP | 70.26±9.46 | 69.98±9.39 | 70.74±9.57 | 0.366 |
| **Ethnicity** | | | | |
| • White European | 481/607 (79.2%) | 303/394 (76.9%) | 178/213 (83.6%) | 0.104 |
| • South Asian | 67/607 (11%) | 46/394 (11.7%) | 21/213 (9.9%) | |
| • Other (Black African/ Caribbean or mixed ethnicity) | 59/607 (9.7%) | 45/394 (11.4%) | 14/213 (6.6%) | |
| **Smoking in pregnancy*** | | | | |
| • Never smoked | 270/583 (46.3%) | 190/373 (50.9%) | 80/210 (38.1%) | <0.001 |
| • Ex-smoker (stopped before or during pregnancy) | 216/583 (37.0%) | 147/373 (39.4%) | 69/210 (32.9%) | |
| • Current smoker | 97/583 (16.6%) | 36/373 (9.7%) | 61/210 (29.0%) | |
| **Alcohol in pregnancy** | | | | |
| • Never drank | 264/567 (46.6%) | 160/361 (43.2%) | 104/206 (50.5%) | 0.098 |
| • Ex-drinker (stopped before or during pregnancy) | 288/567 (50.8%) | 194/361 (53.7%) | 94/206 (45.6%) | |
| • Current drinker | 15/567 (2.6%) | 7/361 (1.9%) | 8/206 (3.9%) | |
| **Marital status** | | | | |
| • Single | 47/572 (8.2%) | 21/366 (5.7%) | 26/206 (12.6%) | 0.004 |
| **Employment status** | | | | |
| • Unemployed | 18/547 (3.3%) | 9/352 (2.6%) | 9/195 (4.6%) | 0.196 |
| Previous GDM pregnancy | 50/607 (8.2%) | 26/394 (6.6%) | 24/213 (11.3%) | 0.545 |
| **At OGTT and Intrapartum** | | | | |
| Gestational age at diagnosis (weeks) | 27.94±4.41 | 28.16±4.21 | 27.53±4.74 | 0.093 |
| Fasting glucose (mmol/l)* | 5.02±0.89 | 4.96± 0.89 | 5.12±0.89 | 0.042 |
| 2hrs glucose (mmol/l)** | 8.41±1.76 | 8.55±1.75 | 8.13±1.75 | 0.008 |
| HbA1c (mmol/mol) | 35.74±5.05 | 35.52±4.69 | 36.17±5.70 | 0.154 |
| Gestational age at birth (weeks) | 37.92±1.32 | 37.91±1.27 | 37.95±1.39 | 0.711 |
| Preterm (GA <37 weeks) | 85/599 (14.2%) | 53/390 (13.6%) | 32/209 (15.3%) | 0.565 |
| **Mode of delivery** | | | | |
| • Spontaneous | 309/600 (51.5%) | 197/391 (50.4%) | 112/209 (53.6%) | 0.688 |
| • Instrument assisted | 50/600 (8.3%) | 32/391 (8.2%) | 18/209 (8.6%) | |
| • Caesarean delivery | 241/600 (40.2%) | 162/391 (41.4%) | 79/209 (37.8%) | |
| Birthweight (gms) | 3209.23±490.39 | 3211.95±467.75 | 3204.14±531.49 | 0.853 |
| **Birth centiles** | | | | |
| • AGA (10-90[th]centile) | 394/551 (71.5%) | 264/356 (74.2%) | 130/195 (66.7%) | 0.058 |
| • SGA (<10 centile) | 81/551 (14.7%) | 43/356(12.1%) | 38/195 (19.5%) | |
| • LGA (>90 centile) | 76/551 (13.8%) | 49/356 (13.8%) | 27/195 (13.8%) | |

*(Continued)*

**Table 1.** (Continued)

| Characteristics | All women $n$ = 607 | Attended $n$ = 394 | Did not attend $n$ = 213 | p-value |
|---|---|---|---|---|
| Male baby n (%)** | 307/599 (51.3%) | 183/390 (46.9%) | 124/209 (59.3%) | 0.004 |
| Breastfeeding initiated** | 293/545 (53.8%) | 207/355 (58.3%) | 86/190 (45.3%) | 0.004 |
| Timing of ppGT (days) | - | 67.63±18.32 | - | |

Values are expressed in mean ± standard deviation and n (%) as appropriate; BW, Birthweight; AGA, Appropriate for gestational age; SGA, Small for gestational age; LGA, Large for gestational age.

GDM women for ppGT (Fig 3). Our proposed ML prediction model has the highest benefit across various probability thresholds and specifically identified 46% GDM women who are unlikely to attend ppGT (optimal threshold = 0.464). Based on our proposed final model, the composite risk score, P (non-attendance to ppGT), was calculated as $1/[1+\exp(-b)]$, where $b$ = −3.1599 + (0.1926×antenatal fasting glucose) + (−0.1415×antenatal postprandial glucose) + (0.0195×antenatal HbA1c) + (−0.0410×gestational age at antenatal OGTT) + (−0.0797×maternal age) + (0.0027×booking BMI) + (0.5486×parity) + (−0.8447×married) + (−0.2392×other ethnicity) + (0.8312×current smoker) + (0.0116×diastolic BP) + (0.1062 ×gestational age at birth) + (0.7418×women delivered SGA infants) + (0.4761×male sex of the baby) + (0.5630×instrumental delivery) + (−0.2115×initiated breastfeeding) (S2 Appendix).

During the subsequent follow-up period between 4 and 24 months post-delivery, two hundred and thirty- three GDM women had at least one HbA1c tested value (Fig 1). Among those tested ($n$ = 233 out of 607), additional 24% ($n$ = 56) women had dysglycaemia. Twelve women had converted to T2D between ppGT and 24 months with higher proportion in women who did not attend the immediate ppGT (attended vs non-attended: 2.5 vs. 11.4%; p = 0.005).

**Table 2. Factors associated with non-attendance to postpartum screening identified by logistic regression.**

| Predictors | β (SE) | OR (95% CI) |
|---|---|---|
| Intercept | -3.1599 (3.337) | - |
| Maternal age | -0.0797 (0.019) | 0.92 (-0.116, -0.043) |
| Antenatal fasting glucose | 0.1926 (0.145) | 1.21 (-0.092, 0.478) |
| Antenatal 2-hrs Glucose | -0.1415 (0.062) | 0.87 (-0.262, -0.021) |
| Antenatal HbA1c | 0.0195 (0.024) | 1.02 (-0.028, -0.067) |
| Gestational age at GDM diagnosis | -0.0410 (0.022) | 0.96 (-0.084, 0.002) |
| Booking BMI | 0.0027 (0.016) | 1.00 (-0.028, 0.033) |
| Continuing to smoke at booking | 0.8312 (0.258) | 2.30 (0.325, 1.338) |
| Unmarried at booking | -0.8447 (0.345) | 0.43 (-1.521, -0.168) |
| Diastolic BP at booking | 0.0116 (0.011) | 1.01 (-0.010, 0.034) |
| Other Ethnicity (Black African/ Caribbean or mixed ethnicity) | -0.2392 (0.351) | 0.79 (-0.928, 0.450) |
| Gestational age at birth | 0.1062 (0.076) | 1.11 (-0.043, 0.255) |
| Instrument assisted delivery | 0.5630 (0.348) | 1.76 (-0.118, 1.244) |
| Women delivered SGA infants | 0.7418 (0.275) | 2.10 (0.204, 1.280) |
| Women delivered male babies | 0.4761 (0.194) | 1.61 (0.096, 0.856) |
| Breastfeeding initiation before discharge | -0.2115 (0.210) | 0.81 (-0.622, 0.199) |
| Parity | 0.5486 (0.096) | 1.73 (0.361, 0.736) |

ML: Machine learning; OR: Odds ratio; 95%CI, confidence interval. Logistic regression model was fit using features selected from lasso by machine learning algorithm.

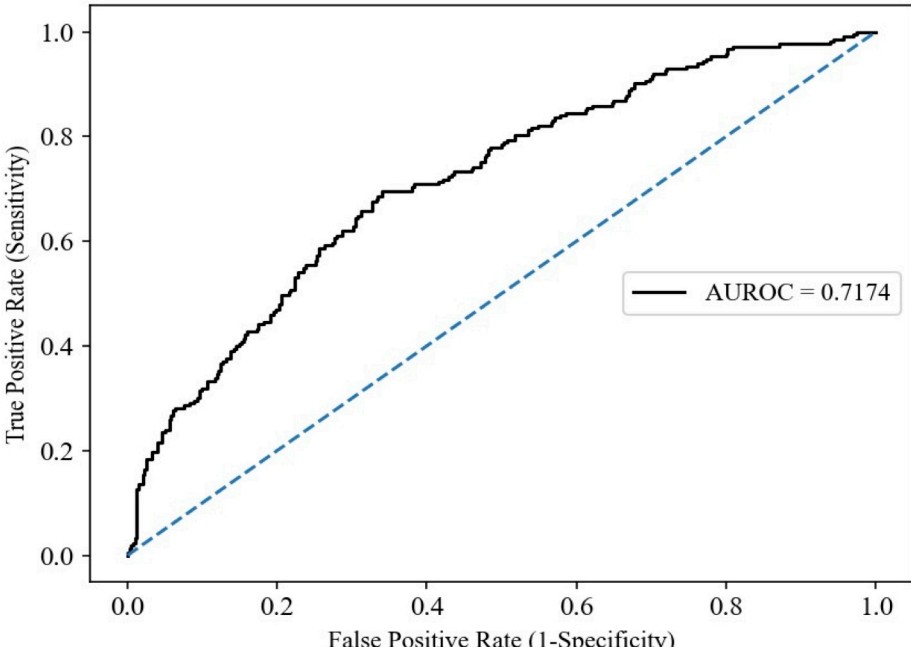

**Fig 2. AUROC for prediction of non-attendance at ppGT.** AUROC was used to evaluate the performance of our machine learning based algorithm using logistic regression model on the validation cohort, n = 607 by aggregating the predictions from the 5 test folds of CV1. The area under ROC was 0.72. The dotted line indicates optimal threshold. The grey line indicates 'target none' approach and black line indicates 'target all' approach. The blue line indicates the net benefit of the proposed ML prediction model.

Survival analysis showed that women from south Asian ethnicity, those with higher booking BMI and antenatal HbA1c had increased hazard ratio for dysglycaemia in both ppGT groups (S1 Table).

## Discussion

Our study highlights that unmarried status, younger age, higher BMI, multiparity, and continued smoking during pregnancy are risk factors of poor postpartum attendance for glucose testing, following a GDM pregnancy. Using an unbiased, data-driven machine learning approach, we propose a composite risk score based on easily available antenatal parameters for women who are less likely to attend postpartum screening. Worryingly, it appears that those who did

**Table 3. Sensitivity and specificity of postpartum glucose attendance by ML algorithm at various probability thresholds.**

| Probability threshold | Sensitivity | Specificity | PPV | NPV | F1 | Accuracy | Proportion attended ppGT |
|---|---|---|---|---|---|---|---|
| 0.27 | 0.90 | 0.32 | 0.42 | 0.85 | 0.57 | 0.52 | 25 |
| 0.36 | 0.80 | 0.48 | 0.45 | 0.82 | 0.58 | 0.59 | 38 |
| 0.39 | 0.75 | 0.53 | 0.46 | 0.80 | 0.57 | 0.60 | 43 |
| 0.46* | 0.70 | 0.66 | 0.52 | 0.80 | 0.60 | 0.67 | 54 |
| 0.53 | 0.60 | 0.72 | 0.54 | 0.77 | 0.57 | 0.68 | 61 |

PPV: Positive predictive value; NPV: Negative predictive value;

* Optimal threshold with maximal F1 score that shows sensitivity of 0.70 and specificity of 0.66 to determine the number of GDM women to be focused for postpartum glucose testing.

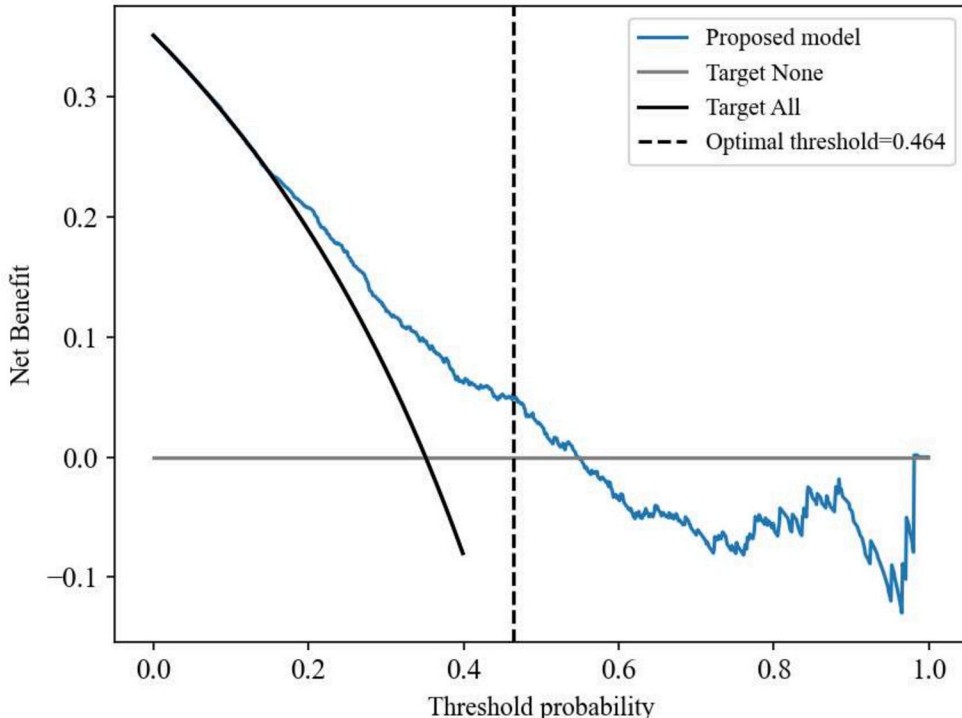

**Fig 3. Decision curve analysis for the standardized net benefit obtained from the proposed ML model.** The DCA (Decision curve analysis) showed the net benefit obtained from the ML (blue line) prediction model compared to the target all (solid black line) or target none (solid grey line). Net benefit by implementing our model in a clinical setting is larger when compared to the follow-up of all GDM women for ppGT. DCA was derived from the equation, Net benefit $= \frac{1}{N}\left[ TP - FP\left(\frac{p_t}{1 - p_t}\right)\right]$, where $TP$ and $FP$ are the true positives and false positives respectively, $p_t$ is the probability threshold, and $N$ is the total number of participants in the validation cohort, $N = 607$.

not attend the immediate postpartum screening, have higher conversion rate to T2D within two years of index pregnancy, although the numbers were small.

The attendance to postpartum glucose tolerance screening in our centre has improved since the introduction of individualised letter at the time of discharge in 2015 (64.9% vs 49%) [10]. Although the uptake is similar [12], or better than others [1, 19], it is still not satisfactory. Interestingly, 33% of women who did not attend the immediate testing, subsequently were tested at least once within two years of index pregnancy. This suggests that the message of annual screening to the patient and/or their GP is filtering through, albeit in a small proportion of individuals. However, we did not have the information on how many women were invited for this subsequent testing.

Previous studies have attempted to understand the barriers for poor attendance. Lack of awareness and competing challenges with childcare are some of the common barriers [14]. In addition, younger women may perceive their risk is low [20]. While obese women may understand their risk, they might not attend for other reasons including fear or the stigma of diagnosis of T2D [21]. Antenatal educational interventions highlighting the subsequent risk of T2D and flexible postnatal lifestyle services by incorporating health visitors for glucose testing have been suggested as potential strategies to improve these barriers [13, 22]. Although implementation of a recall register system and/or coordinator institution enhanced the ppGT uptake rate in some countries, our ML based risk stratification could provide a better understanding for personalised education specifically, to capture non-attenders at the time of hospital discharge

[11, 12]. While our letter is designed to have individualised approach, and seem to have improved the overall testing, it is possible that some women felt the information provided is standard and is 'not specific to them'.

Our proposed machine learning based individualised probability threshold identified 46% of women who are unlikely to attend postpartum testing using simple, routinely available characteristics. Healthcare professionals can identify and target these women antenatally and provide enhanced education on the importance of the postpartum glucose testing both during pregnancy and at the time of discharge from the hospital. This approach may also facilitate healthcare professionals' perception of ppGT by emphasizing the significance of screening that may help women to identify their risk and seek professional care for timely intervention. Alternatively, doing FPG just before discharge could also be carried out in these women, to ensure some form of glucose testing is done [23]. To our knowledge this is the first study to attempt to create an individualised composite risk score for postpartum non-attendance, using a machine learning algorithm. A simple, Microsoft Excel based individualised risk score can be easily calculated by the healthcare professionals to identify the women who is less likely to attend that may benefit from enhanced education (S2 Appendix).

The overall conversion rate to abnormal glucose tolerance in our study was high i.e., 24.4% at the ppGT and an additional 24% in the subsequent 18 months. This is worryingly higher than previous reports [1, 3], and highlights the importance of ensuring testing happens not only in immediate post-delivery, but annually thereafter. Recent evidence also suggests that the cardiovascular risk is higher in these women [4, 22] and, perhaps the annual screening should include other common CVD risk factors such as smoking, blood pressure and lipid profile. Individualised risk calculation for abnormal glucose tolerance similar to the one proposed for non-attendance could potentially improve attendance, utilize the resources effectively and enable targeted education for prevention. As our composite risk score can be calculated in a simple Excel based calculator (S2 Appendix) our approach can also be easily implemented in low resource settings.

Our study had two key strengths: follow-up data on women who did not attend the immediate post-partum testing and machine learning based individualised composite risk score for non-attendance. However, we note that although this work included all the women who were diagnosed to have GDM in the study period, this was a retrospective study. We therefore were limited by the routinely collected data. In addition to some missing data, we did not have information about treatment modalities, which has been shown to be associated with postpartum screening uptake, albeit with mixed results [24–26].

Other factors could conceivably be relevant in predicting women's participation in follow-up, for example, proximity of the healthcare centre, their employment status and the nature of their employment, flexibility in having an appointment. However, in this study, all participants were proximal to the hospital, had their antenatal care and delivery in the hospital, women were able to reschedule and pick appointment date/time that is suitable for them. We also found no statistical difference between employed and unemployed women. But these and other socioeconomic parameters may prove relevant in larger studies.

While this information might have improved the predictive power of the ML algorithm, we believe that using our model it is easy to incorporate additional maternal characteristics based on their availability. We also did not have information on how many women were invited for the subsequent annual testing. Finally, while we are confident about the beneficial role of advanced nurse practitioner for the improvement in our postpartum testing, we were unable to ascertain the role of the standard letter sent to the women.

## Conclusion

Our study highlights that a simple machine learning algorithm can accurately identify women who are unlikely to attend postpartum testing, based on routinely collected clinical parameters. Such knowledge can enable healthcare professionals to provide enhanced education throughout the antenatal period, at the time of discharge, and during the immediate postpartum period using health visitors. However, this will require an RCT to confirm the usefulness of ML based risk prediction in the evaluation of postpartum glucose screening uptake rate. In addition, with the improved accuracy of point of care HbA1c testing kits, this can also be utilized to improve the testing and identification of abnormal glucose tolerance [27]. The cost effectiveness of such strategies will also require well designed prospective studies.

## Supporting information

**S1 Fig. Mean CV-accuracy as a function of lasso regularization hyperparameter C.** S1 Fig shows the variation in the mean stratified 10-fold cross validation accuracy as a function of the lasso regularization hyperparameter C for the final model. Maximum CV-accuracy of 0.7065 is obtained for C = 0.1325.
(TIF)

**S2 Fig. Concept diagram of the proposed ML algorithm.** The processed data is divided into 5 folds. The light grey region shows the training data, and the dark grey shows the testing data. Within each iteration i of the outer 5-fold CV, the training folds further undergo internal 10-fold CV in Step 1 for lasso hyperparameter optimization. This is known as nested cross validation. In step 2, feature selection is performed on the training folds in iteration i using lasso with optimized regularization hyperparameter. Logistic regression model with selected features is fit on the training folds in iteration i in Step 3. The fit model is used for prediction on the exclusively held out test data in iteration i. The test predictions from all 5 iterations are aggregated to plot and calculate the area under the ROC curve for evaluating the performance of our method.
(TIF)

**S3 Fig. Feature selection using lasso regularization in each of the 5 iterations of CV1 and the final model.** S3 Fig gives a visual understanding of the features selected in each iteration of CV1 and as well as in the final model. Pink cells show the selected features in CV1, green cells show the selected features in the final model built on the full data and red cells show the feature coefficients shrank to zero in both CV1 and the final model. The sets of features selected in all folds of CV1 (except fold 3, where all features except two are selected) and those selected in the final model are similar. The number of samples in the training and testing folds, value of the optimized hyperparameter C, number of features selected and area under the ROC curve for the test fold for each iteration i of CV1 as well as for the final model are provided below each respective column.
(TIF)

**S4 Fig. Comparison of correct prediction of non-attendance versus follow-ups avoided for different probability thresholds using ML proposed model.** S4 Fig shows the comparison of correct prediction of non-attendance versus follow-ups avoided for different probability thresholds using our proposed model. Based on our proposed final model, the composite risk score, P(non-attendance), is calculated as 1/1+exp(-b), where b = -3.1599 + (0.1926*antenatal fasting glucose) + (-0.1415*antenatal postprandial glucose) + (0.0195*antenatal HbA1c) + (-0.0410*gestational age at antenatal OGTT) + (-0.0797*maternal age) + (0.0027*booking

BMI) + (0.5486*parity) + (-0.8447*married) + (-0.2392*other ethnicity) + (0.8312*current smoker) + (0.0116*diastolic BP) + (0.1062*gestational age at birth) + (0.7418*women delivered SGA infants) + (0.4761*male sex of the baby) + (0.5630*instrumental delivery) + (-0.2115*initiated breastfeeding). SGA baby, smoker, marital status, male sex of baby, and other ethnicity are binary variables, taking value of either 0 or 1 depending on their absence or presence respectively.
(TIF)

**S5 Fig. F1 graph of optimal probability threshold.** F1 graph showing the model sensitivity and specificity at variable thresholds.
(TIF)

**S1 Table. Cox proportional hazard ratio for dysglycaemia within 2 years of index pregnancy.**
(DOCX)

**S1 Appendix. Standard letter format for postpartum screening to all GDM women issued from the Diabetes clinic, GEH-NHS Trust, UK.**
(PDF)

**S2 Appendix. Microsoft Excel based risk stratification of ppGT non-attendance in GDM women.**
(XLSX)

**S3 Appendix. Supporting information file consisting minimal anonymized dataset of postpartum attendance in GDM women.** Full dataset is available on request following completion of suitable confidentiality agreement.
(CSV)

## Acknowledgments

This study was conducted as an audit at the George Eliot Hospital NHS Trust, Nuneaton, UK. GEH NHS Trust had no role in the design, data collection, analysis, reporting and interpretation of the data. Preliminary findings of this study were presented at the 56th Annual Meeting of the European Association for the study of Diabetes (EASD) in 21-25[th] September 2020. We extend our thanks to Mrs J Wilson, Mrs J Plester, Mrs T Ritchie and Mr S Selvamoni for their help in providing the list of all the GDM women. Mrs T Ritchie is the advanced nurse practitioner who facilitates the postpartum testing.

## Author Contributions

**Conceptualization:** Nishanthi Periyathambi, Yonas Ghebremichael-Weldeselassie, Vinod Patel, Nithya Sukumar, Ponnusamy Saravanan.

**Data curation:** Nishanthi Periyathambi.

**Formal analysis:** Nishanthi Periyathambi, Durga Parkhi.

**Funding acquisition:** Ponnusamy Saravanan.

**Methodology:** Nishanthi Periyathambi, Durga Parkhi, Yonas Ghebremichael-Weldeselassie, Nithya Sukumar, Rahul Siddharthan, Leelavati Narlikar, Ponnusamy Saravanan.

**Project administration:** Ponnusamy Saravanan.

**Resources:** Vinod Patel, Nithya Sukumar, Ponnusamy Saravanan.

**Software:** Nishanthi Periyathambi, Durga Parkhi, Yonas Ghebremichael-Weldeselassie, Rahul Siddharthan, Leelavati Narlikar.

**Supervision:** Nithya Sukumar, Ponnusamy Saravanan.

**Validation:** Nishanthi Periyathambi, Durga Parkhi, Yonas Ghebremichael-Weldeselassie, Nithya Sukumar, Rahul Siddharthan, Leelavati Narlikar.

**Visualization:** Nishanthi Periyathambi, Durga Parkhi, Rahul Siddharthan, Leelavati Narlikar.

**Writing – original draft:** Nishanthi Periyathambi.

**Writing – review & editing:** Nishanthi Periyathambi, Durga Parkhi, Yonas Ghebremichael-Weldeselassie, Vinod Patel, Nithya Sukumar, Rahul Siddharthan, Leelavati Narlikar, Ponnusamy Saravanan.

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
