## [Decision Letter · Decision Letter 0]

15 Dec 2021

PONE-D-21-35154Machine learning prediction of non-attendance to postpartum glucose screening and subsequent risk of type 2 diabetes following gestational diabetesPLOS ONE

Dear Dr. Saravanan,

Thank you for submitting your manuscript to PLOS ONE. After careful consideration, we feel that it has merit but does not fully meet PLOS ONE’s publication criteria as it currently stands. Therefore, we invite you to submit a revised version of the manuscript that addresses the points raised during the review process.

We look forward to receiving your revised manuscript.

Kind regards,

Rajakumar Anbazhagan, Ph. D

Academic Editor

PLOS ONE

Journal Requirements:

 [PS and YW are part funded by Medical Research Council, UK Grant number:  MR/R020981/1 Funder: MRC- UK URL: https://mrc.ukri.org/funding/

NP is funded by Chancellor's international Scholarship for doctoral research.

Funder: University of Warwick

URL: https://warwick.ac.uk/services/dc/schols_fund/scholarships_and_funding/chancellors_int/

The funders had no role in study design, data collection and analysis, decision to publish, or preparation of the manuscript.].

3. We noted in your submission details that a portion of your manuscript may have been presented or published elsewhere. [A subset of findings were presented as abstract at the 56th Annual Meeting of the European Association for the study of Diabetes (EASD) in 21-25th September 2020. No data tables and/or figures from this manuscript were published or presented elsewhere.] Please clarify whether this conference proceeding was peer-reviewed and formally published. If this work was previously peer-reviewed and published, in the cover letter please provide the reason that this work does not constitute dual publication and should be included in the current manuscript.

Reviewers' comments:

Reviewer's Responses to Questions

**Comments to the Author**

1. Is the manuscript technically sound, and do the data support the conclusions?

Reviewer #1: Yes

Reviewer #2: Yes

Reviewer #3: Yes

Reviewer #4: Yes

2. Has the statistical analysis been performed appropriately and rigorously? 

Reviewer #1: Yes

Reviewer #2: Yes

Reviewer #3: I Don't Know

Reviewer #4: Yes

3. Have the authors made all data underlying the findings in their manuscript fully available?

Reviewer #1: Yes

Reviewer #2: Yes

Reviewer #3: Yes

Reviewer #4: Yes

4. Is the manuscript presented in an intelligible fashion and written in standard English?

Reviewer #1: Yes

Reviewer #2: Yes

Reviewer #3: Yes

Reviewer #4: Yes

5. Review Comments to the Author

Reviewer #1: The manuscript authored by Nishanthi Periyathambi et al aimed to identify the factors associated with non attendance of postpartum glucose test by developing a machine leaning algorithm using available information on a cohort of gestational diabetes mellitus women (607). Their algorithm was able to predict which individuals are unlikely to attend postpartum glucose testing. The study is particularly interesting because it gives an opportunity to the health care system to adapt their strategy to improve the follow up of these individuals using simple tools.

The manuscript is well written and the data are supporting the conclusion. There is minor comments to improve the paper:

1/ It would be very helpful to further extend the discussion to include what other factors the current model is not taking in consideration and which may be very interesting to improve the accuracy of the prediction (My first thought goes to the proximity to a testing center, the flexibility in having an appointment, the nature of jobs that these individuals have...etc . How this can be addressed in future studies,...etc.

2/ The font is not homogenous in the discussion section and need to be adjusted.

Reviewer #2: In this manuscript by Nishanthi Periyathambi et al., describe factors who did not attend gestational type 2 diabetes (T2D) screening in the postpartum period using machine learning algorithms. This study is aimed to identify patient characteristics who did not attend the immediate ppGT and assess their subsequent T2D risk. Authors built a predictive model using machine learning algorithms and proposed factors associated with non-attendance of immediate postpartum glucose test. Further authors think, improved personalised education may improve postpartum glucose screening. Overall factors associated with the gestational T2D, are interesting. I have a few questions on this study, authors have to address these following questions.

Major comments

- Authors should highlight significant values in tables with asterisk marks or underline etc., for younger, unmarried, multiparous, BMI and smoking during pregnancy.

- Authors should describe the details of model analysis in the methods section. Readers benefited by including the S1 file which mentioned "Stepwise 1 description of the Machine learning" into methods section.

- Results sections of the study were not well explained, what is the concept diagram used in this study, how does it generate helpful outcomes of the figure S2 for next steps. The area under ROC was 0.72. How does this value help to decide the factors associated with non attendance of glucose tests? The outcome results of decision curve analysis is not explained in detail. How does decision curve analysis can be used to find factors or glucose levels in women who did not attend gestational type 2 diabetes (T2D) screening in the postpartum period.

- Authors should provide any previous examples or references for the machine learning (ML) used in this study. How does this ML help in the example disease predictive models?

- Authors mention in line 148 “After getting assurance of acceptable performance of this method” how does the assurance of method was confirmed by authors in this study, please explain features that help to get confidence of the model.

Minor comments

- Authors should report how hospitals can be improved the personalised education for postpartum glucose screening using this study, whether this can be applied to hospitals in many countries or specific to a country. How it can be improved could be described in discussion or if any country has better screening for gestational type 2 diabetes (T2D) in the postpartum period, additional details of these elements could be helpful to improve the study or personalised education.

- Add a short form for the area under the ROC as AUROC for figure2 legend or results section describe figure2.

- Label x and y axis in fig S5.

Reviewer #3: Reviewer’s comment

In the study titled ‘Machine learning prediction of nonattendance to postpartum glucose screening and subsequent risk of type 2 diabetes following gestational diabetes,’ the authors Periyathambi et al. have attempted to find the characteristics/habits of the women who did not attend postpartum glucose testing (ppGT) despite the reminder from the hospital staff. This is a retrospective cohort study of all 607 GDM women who were ppGT due between January 2016 and December 2019 at the George Eliot Hospital NHS Trust, UK. This study indicates that the women who did not attend ppGT were younger, multiparous, obese, and continued smoking during pregnancy. Most of these characteristics of individuals are comparable with their previous study (Venkararaman et al., 2015). The authors have used machine learning (Python version 3.7) for predicting the type of women who were more likely to have nonattendance for ppGT. This study has many limitations such as small sample size, uniformity in reminding the individuals, etc. However, this preliminary study can encourage researchers to do a study on a larger sample size in the future. Following are a few suggestions that might improve the article.

1. The authors may explain how different machine learning results are compared to other analyses such as manual analysis.

2. The in-person consultation with hospital staff at the time of discharging from the hospital after delivery can play an important role in whether the woman will attend ppGT or not. Also, the amount of importance given for the consultation may vary among hospital staff.

3. The authors may consider including a future direction such as ‘A study with larger sample size would give better clarity on machine learning for nonattendance for ppGT.

Reviewer #4: The manuscript by Periyathambi et al, titled " Machine learning prediction of non-attendance to postpartum glucose screening and subsequent risk of type 2 diabetes following gestational diabetes" studies the social and physiological factors associated with non-attendance of immediate postpartum glucose test using machine learning algorithm in women with gestational diabetes mellitus.

The study also presents correlation between non-attendance and higher risk of conversion to type 2 diabetes mellitus.

The study is well designed and executed and the import of the report is evident in its findings. However, I would like the authors to discuss the effectiveness of their machine learning model and its predictive power in a different socio-economic setting – for example in under-developed or developing societies. This would certainly enhance the quality and relevance of the manuscript.

Overall, the authors did an excellent job in organizing their workflow, and the manuscript is easily understandable even for non-expert readers. I applaud the authors for their commendable effort.

6. PLOS authors have the option to publish the peer review history of their article (what does this mean?). If published, this will include your full peer review and any attached files.

Reviewer #1: No

Reviewer #2: No

Reviewer #3: **Yes: **Anil Mathew Tharappel

Reviewer #4: **Yes: **Madhurima Dhara

---

## [Author Response · Author response to Decision Letter 0]

7 Feb 2022

PONE-D-21-35154

Machine learning prediction of non-attendance to postpartum glucose screening and subsequent risk of type 2 diabetes following gestational diabetes

Dear Reviewers,

REVIEWER 1 COMMENTS:

1. It would be very helpful to further extend the discussion to include what other factors the current model is not taking in consideration, and which may be very interesting to improve the accuracy of the prediction (My first thought goes to the proximity to a testing center, the flexibility in having an appointment, the nature of jobs that these individuals have...etc. How this can be addressed in future studies, etc.

Response:

We thank the reviewers for this important point. 

· We agree that factors such as access to nearest healthcare centre and travel distance could impact women’s participation in screening (Cullinan et al, 2011). However, all the study participants in our cohort had easy access to the testing centre (the catchment area is small, less than 5 miles for those who live in the city and easy/free parking at the hospital). They all are given the option to choose their appointments too if the offered appointment date/time is not suitable. We acknowledge that our anonymous data collection process could not incorporate otherwise identifiable records, i.e personal address and postcode. 

· We have supplemented the S1 Appendix section of the invite that implies flexibility to have an appointment at the client’s earliest availability. In addition, all the participants had telephone reminders regarding their appointment booking by one of the specialist nurse practitioners where it provides an opportunity for the women to amend their screening date and/or time. 

· Although we found higher proportion of employed women (64.8%) attending the postpartum glucose testing compared to unemployed women (50%), it was not statistically significant. We have also incorporated our clarification in discussion in page 15, lines 335:341. We agree further research on the barriers and facilitators of postpartum glucose testing is needed to provide additional insights.

2. The font is not homogenous in the discussion section and need to be adjusted.

Response:

We have addressed this comment by modifying the font size to 11pts in page, 11:12 and a spacing in line 318 on page 14.

REVIEWER 2 COMMENTS:

3. Authors should highlight significant values in tables with asterisk marks or underline etc., for younger, unmarried, multiparous, BMI and smoking during pregnancy.

Response:

Thank you for this suggestion. We have incorporated this comment by including a separate column for p-values and asterisk marks in the ‘Characteristics’ column to show significance in Table.1 in ‘Revised Manuscript with Track Changes’.

4. Authors should describe the details of model analysis in the methods section. Readers benefited by including the S1 file which mentioned "Stepwise 1 description of the Machine learning" into methods section.

Response:

We agree with your comments and have incorporated this suggestion in the Method section in page 6:7, lines 150:168.

5. a) Results sections of the study were not well explained, what is the concept diagram used in this study, how does it generate helpful outcomes of the Figure S2 for next steps. 

b) The area under ROC was 0.72. How does this value help to decide the factors associated with non-attendance of glucose tests? 

c) The outcome results of decision curve analysis is not explained in detail. 

d) How does decision curve analysis can be used to find factors or glucose levels in women who did not attend gestational type 2 diabetes (T2D) screening in the postpartum period.

Response: 

a) We would like to clarify how the concept diagram used in this study as follows: The processed data is divided into 5 folds. The light grey region shows the training data, and the dark grey shows the testing data. Within each iteration i of the outer 5-fold CV, the training folds further undergo internal 10-fold CV in Step 1 for lasso hyperparameter optimization. This is known as nested cross validation. In step 2, feature selection is performed on the training folds in iteration i using lasso with optimized regularization hyperparameter. Logistic regression model with selected features is fit on the training folds in iteration i in Step 3. The fit model is used for prediction on the exclusively held out test data in iteration i. The test predictions from all 5 iterations are aggregated to plot and calculate the area under the ROC curve for evaluating the performance of our method. We have updated the figure caption to explain this.

b) A non-informative classifier which picks the label at random, will have a 0.5 area under the ROC (AUROC), while a perfect one will have an AUROC of 1.0. A value of 0.72 on an unseen (test) dataset denotes that the classifier is informative, although not perfect. Different classifiers will result in different areas, the 0.72 was the best possible area among the methods we tried. The factors that give the highest area under the curve on an unseen dataset are listed as the most prominent factors associated with non-attendance of glucose tests. 

c) We have redrafted the result section by incorporating decision curve analysis (DCA) outcome with the optimal threshold (0.464) to target non-attenders based on our prediction model in page 12, lines 248: 250 to be more in line with your comments. The DCA has higher net benefit than the conventional strategy to focus all the women or targeting none which would lead to improved glucose screening rate in any clinical settings. We have also updated the Fig.3 legend to show the DCA comparisons between the prediction model vs ‘target all’ or ‘target none’ lines in page 13, lines 260:261. 

d) The DCA evaluated our prediction model which encompasses all the variables that were used to construct the final model (antenatal OGTT values, i.e, fasting glucose and 2-hrs glucose and gestational age at the time of screening, antenatal HbA1c, age, booking BMI, parity, marital status, ethnicity, smoking status, booking BP, gestational age at birth, infant birth size and sex, type of delivery and breastfeeding initiation assessed before the hospital discharge). Therefore, our DCA has overcome the limitation of discriminating women who would not have been screened for gestational diabetes and otherwise targeted via other predictive factors. We would like to further address your comments by highlighting the S2 Appendix, an Excel based calculator which shows the combination of all the predictive factors to identify women who are less likely to attend the ppGT. In simple terms, DCA was not implemented to find factors associated with non-attendance; DCA helps to understand that the classifier built with the chosen factors (corresponding to highest AUROC) is more beneficial than follow-up of all GDM women assuming none will attend.

6. Authors should provide any previous examples or references for the machine learning (ML) used in this study. How does this ML help in the example disease predictive models?

Response: 

Thank you for your suggestion. We have included a reference for machine learning approach to predict a disease (Type 2 Diabetes after GDM pregnancy) in page, 6, line number 134. However, while the basic methods such as logistic regression and lasso are standard, the detailed framework as presented here is our own, and approaches like nested cross validation to find optimal hyper-parameters have not been used before in such studies, to the best of our knowledge.

7. Authors mention in line 148 “After getting assurance of acceptable performance of this method” how does the assurance of method was confirmed by authors in this study, please explain features that help to get confidence of the model.

Response:

The term “acceptable performance” refers to the AUROC of 0.72 over the unseen data. A value consistently higher than 0.7 i.e., better than the 0.5 a random classifier would achieve assures us of its performance. Along with AUROC, high sensitivity as well as specificity together help to get confidence from the model.

8. Authors should report how hospitals can be improved the personalised education for postpartum glucose screening using this study, whether this can be applied to hospitals in many countries or specific to a country. How it can be improved could be described in discussion or if any country has better screening for gestational type 2 diabetes (T2D) in the postpartum period, additional details of these elements could be helpful to improve the study or personalised education.

Response: 

We have elaborated the usefulness of our ML predictive model in better understanding of personalised education that could improve the ppGT uptake rate in page 14, lines 298:301. We hope these revisions provide a more balanced discussion on the implementation of individualised education in addition to the reminder or recall systems described in few previous studies, Benhalima et al (2017) and Carmody et al (2015).

9. Add a short form for the area under the ROC as AUROC for figure2 legend or results section describe figure2.

Response:

We have addressed your suggestion in page 6 in line 145 and in page 11 in line 225 to rewrite the short form of area under the ROC. 

10. Label x and y axis in fig S5.

Response:

We have addressed this comment by including ‘x’ and ‘y’ axis label as ‘Precision’ and ‘Recall’ for the S5 Fig.

REVIEWER 3 COMMENTS:

11. The authors may explain how different machine learning results are compared to other analyses such as manual analysis.

Response:

We are unsure of what the reviewer means by manual analysis. We assume the reviewer means standard statistical analysis. ML methods used here are of the predictive kind, where the eventual goal is to predict the class of a new case (not seen thus far by the model). Statistical analyses will identify differences across individual predictors, between the classes (the p-value listed in Table.1), but do not give a score or a prediction based on them. We note that the final formula learned by the method, that does the best among all methods we tried, can in fact be applied using a simple calculator (as provided in the S2 Appendix) by the healthcare professional at his/her own clinic. 

12. The in-person consultation with hospital staff at the time of discharging from the hospital after delivery can play an important role in whether the woman will attend ppGT or not. Also, the amount of importance given for the consultation may vary among hospital staff.

Response:

We agree with you and have incorporated this suggestion in page 14, lines 309:311. 

13. The authors may consider including a future direction such as ‘A study with larger sample size would give better clarity on machine learning for nonattendance for ppGT.

Response:

Thank you for your suggestion and we have revised the text in page 15, lines 353:355 to address the usefulness of ML prediction model.

REVIEWER 4 COMMENTS:

14. The study is well designed and executed, and the import of the report is evident in its findings. However, I would like the authors to discuss the effectiveness of their machine learning model and its predictive power in a different socio-economic setting – for example in under-developed or developing societies. This would certainly enhance the quality and relevance of the manuscript.

Response:

We have incorporated your comments in page 14:15, lines 325:327 to reflect the easily identifiable factors for the evaluation of non-attenders for postpartum screening. Our Excel based calculator- S2 Appendix could be used in any low resource setting to target women who are less likely to turn up for the postpartum screening.

---

## [Decision Letter · Decision Letter 1]

15 Feb 2022

Machine learning prediction of non-attendance to postpartum glucose screening and subsequent risk of type 2 diabetes following gestational diabetes

PONE-D-21-35154R1

Dear Dr. Saravanan,

We’re pleased to inform you that your manuscript has been judged scientifically suitable for publication and will be formally accepted for publication once it meets all outstanding technical requirements.

Kind regards,

Academic Editor

PLOS ONE

Additional Editor Comments (optional):

Reviewers' comments:

Reviewer's Responses to Questions

**Comments to the Author**

1. If the authors have adequately addressed your comments raised in a previous round of review and you feel that this manuscript is now acceptable for publication, you may indicate that here to bypass the “Comments to the Author” section, enter your conflict of interest statement in the “Confidential to Editor” section, and submit your "Accept" recommendation.

Reviewer #2: All comments have been addressed

Reviewer #3: All comments have been addressed

2. Is the manuscript technically sound, and do the data support the conclusions?

Reviewer #2: Yes

Reviewer #3: Yes

3. Has the statistical analysis been performed appropriately and rigorously? 

Reviewer #2: Yes

Reviewer #3: Yes

4. Have the authors made all data underlying the findings in their manuscript fully available?

Reviewer #2: Yes

Reviewer #3: Yes

5. Is the manuscript presented in an intelligible fashion and written in standard English?

Reviewer #2: No

Reviewer #3: Yes

6. Review Comments to the Author

Reviewer #2: All the comments were addressed by the authors.

Reviewer #3: In the revised article, titled 'Machine learning prediction of non-attendance to postpartum glucose screening and

subsequent risk of type 2 diabetes following gestational diabetes' the authors response and changes made are satisfactory.

7. PLOS authors have the option to publish the peer review history of their article (what does this mean?). If published, this will include your full peer review and any attached files.

Reviewer #2: No

Reviewer #3: No

---

## [Editor Report · Acceptance letter]

24 Feb 2022

PONE-D-21-35154R1 

Machine learning prediction of non-attendance to postpartum glucose screening and subsequent risk of type 2 diabetes following gestational diabetes 

Dear Dr. Saravanan:

I'm pleased to inform you that your manuscript has been deemed suitable for publication in PLOS ONE. Congratulations! Your manuscript is now with our production department. 

Kind regards, 

on behalf of

Dr. Rajakumar Anbazhagan 

Academic Editor

PLOS ONE